# Orbit Regularization

**Renato Negrinho**
Instituto de Telecomunicações
Instituto Superior Técnico
1049–001 Lisboa, Portugal
renato.negrinho@gmail.com

**André F. T. Martins**[*]
Instituto de Telecomunicações
Instituto Superior Técnico
1049–001 Lisboa, Portugal
atm@priberam.pt

## Abstract

We propose a general framework for regularization based on group-induced majorization. In this framework, a group is defined to act on the parameter space and an orbit is fixed; to control complexity, the model parameters are confined to the convex hull of this orbit (the orbitope). We recover several well-known regularizers as particular cases, and reveal a connection between the hyperoctahedral group and the recently proposed sorted $\ell_1$-norm. We derive the properties a group must satisfy for being amenable to optimization with conditional and projected gradient algorithms. Finally, we suggest a continuation strategy for orbit exploration, presenting simulation results for the symmetric and hyperoctahedral groups.

## 1 Introduction

The main motivation behind current sparse estimation methods and regularized empirical risk minimization is the principle of *parsimony*, which states that simple explanations should be preferred over complex ones. Traditionally, this has been done by defining a function $\Omega : V \to \mathbb{R}$ that evaluates the complexity of a model $\boldsymbol{w} \in V$ and trades off this quantity with a data-dependent term. The penalty function $\Omega$ is often designed to be a convex surrogate of an otherwise non-tractable quantity, a strategy which has led to important achievements in sparse regression [1], compressed sensing [2], and matrix completion [3], allowing to successfully recover parameters from highly incomplete information. Prior knowledge about the structure of the variables and the intended sparsity pattern, when available, can be taken into account when designing $\Omega$ via sparsity-inducing norms [4]. Performance bounds under different regimes have been established theoretically [5, 6], contributing to a better understanding of the success and failure modes of these techniques.

In this paper, we introduce a new way to characterize the complexity of a model via the concept of *group-induced majorization*. Rather than regarding complexity in an absolute manner via $\Omega$, we define it *relative to a prototype model* $\boldsymbol{v} \in V$, by requiring that the estimated model $\boldsymbol{w}$ satisfies

$$\boldsymbol{w} \preceq_G \boldsymbol{v}, \tag{1}$$

where $\preceq_G$ is an ordering relation on $V$ induced by a group $G$. This idea is rooted in majorization theory, a well-established field [7, 8] which, to the best of our knowledge, has never been applied to machine learning. We therefore review these concepts in §2, where we show that this formulation subsumes several well-known regularizers and motivates new ones. Then, in §3, we introduce two important properties of groups that serve as building blocks for the rest of the paper: the notions of *matching function* and *region cones*. In §4, we apply these tools to the permutation and signed permutation groups, unveiling connections with the recent *sorted $\ell_1$-norm* [9] as a byproduct. In §5 we turn to algorithmic considerations, pinpointing the group-specific operations that make a group amenable to optimization with conditional and projected gradient algorithms.

---

[*]Also at Priberam Labs, Alameda D. Afonso Henriques, 41 - 2°, 1000–123, Lisboa, Portugal.

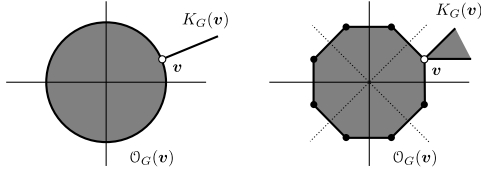

**Figure 1:** Examples of orbitopes for the orthogonal group $\mathsf{O}(d)$ (left) and the hyperoctahedral group $\mathcal{P}_{\pm}$ (right). Shown are also the corresponding region cones, which in the case of $\mathsf{O}(d)$ degenerates into a ray.

A key aspect of our framework is a *decoupling* in which the group $G$ captures the invariances of the regularizer, while the data-dependent term is optimized in the group orbitopes. In §6, we build on this intuition to propose a simple continuation algorithm for orbit exploration. Finally, §7 shows some simulation results, and we conclude in §8.

## 2 Orbitopes and Majorization

### 2.1 Vector Spaces and Groups

Let $V$ be a vector space with an inner product $\langle \cdot, \cdot \rangle$. We will be mostly concerned with the case where $V = \mathbb{R}^d$, *i.e.*, the $d$-dimensional real Euclidean space, but some of the concepts introduced here generalize to arbitrary Hilbert spaces. A *group* is a set $G$ endowed with an operation $\cdot : G \times G \to G$ satisfying *closure* ($g \cdot h \in G$, $\forall g, h \in G$), *associativity* ($(f \cdot g) \cdot h = f \cdot (g \cdot h)$, $\forall f, g, h \in G$), *existence of identity* ($\exists 1_G \in G$ such that $1_G \cdot g = g \cdot 1_G = g$, $\forall g \in G$), and *existence of inverses* (each $g \in G$ has an inverse $g^{-1} \in G$ such that $g \cdot g^{-1} = g^{-1} \cdot g = 1_G$). Throughout, we use boldface letters $\boldsymbol{u}, \boldsymbol{v}, \boldsymbol{w}, \ldots$ for vectors, and $g, h, \ldots$ for group elements. We also omit the group operation symbol, writing $gh$ instead of $g \cdot h$.

### 2.2 Group Actions, Orbits, and Orbitopes

A (left) *group action* of $G$ on $V$ [10] is a function $\psi : G \times V \to V$ satisfying $\psi(g, \psi(h, \boldsymbol{v})) = \psi(g \cdot h, \boldsymbol{v})$ and $\psi(1_G, \boldsymbol{v}) = \boldsymbol{v}$ for all $g, h \in G$ and $\boldsymbol{v} \in V$. When the action is clear from the context, we omit the letter $\psi$, writing simply $g\boldsymbol{v}$ for the action of the group element $g$ on $\boldsymbol{v}$, instead of $\psi(g, \boldsymbol{v})$. In this paper, we always assume our actions are *linear*, *i.e.*, $g(c_1 \boldsymbol{v}_1 + c_2 \boldsymbol{v}_2) = c_1 g\boldsymbol{v}_1 + c_2 g\boldsymbol{v}_2$ for scalars $c_1$ and $c_2$ and vectors $\boldsymbol{v}_1$ and $\boldsymbol{v}_2$. In some cases, we also assume they are *norm-preserving*, *i.e.*, $\|g\boldsymbol{v}\| = \|\boldsymbol{v}\|$ for any $g \in G$ and $\boldsymbol{v} \in V$. When $V = \mathbb{R}^d$, we may regard the groups underlying these actions as subgroups of the *general linear group* $\mathsf{GL}(d)$ and of the *orthogonal group* $\mathsf{O}(d)$, respectively. $\mathsf{GL}(d)$ is the set of $d$-by-$d$ invertible matrices, and $\mathsf{O}(d)$ the set of $d$-by-$d$ orthogonal matrices $\{U \in \mathbb{R}^{d \times d} \mid U^\top U = UU^\top = I_d\}$, where $I_d$ denotes the $d$-dimensional identity matrix.

A group action defines an equivalence relation on $V$, namely $\boldsymbol{w} \equiv \boldsymbol{v}$ iff there is $g \in G$ such that $\boldsymbol{w} = g\boldsymbol{v}$. The *orbit* of a vector $\boldsymbol{v} \in V$ under the action of $G$ is the set $G\boldsymbol{v} := \{g\boldsymbol{v} \mid g \in G\}$, *i.e.*, the vectors that result from acting on $\boldsymbol{v}$ with some element of $G$. Its convex hull is called the *orbitope*:

$$\mathcal{O}_G(\boldsymbol{v}) := \mathbf{conv}(G\boldsymbol{v}). \tag{2}$$

Fig. 1 (left) illustrates this concept for the orthogonal group in $\mathbb{R}^2$. An important concept associated with group actions and orbitopes is that of $G$-majorization [7]:

**Definition 1** *Let $\boldsymbol{v}, \boldsymbol{w} \in V$. We say that $\boldsymbol{w}$ is $G$-majorized by $\boldsymbol{v}$, denoted $\boldsymbol{w} \preceq_G \boldsymbol{v}$, if $\boldsymbol{w} \in \mathcal{O}_G(\boldsymbol{v})$.*

**Proposition 2** *If the group action is linear, then $\preceq_G$ is reflexive and transitive,* i.e., *it is a* pre-order.

*Proof:* See supplemental material. ∎

Group majorization plays an important role in the area of multivariate inequalities in statistics [11]. In this paper, we use this concept for representing model complexity, as described next.

### 2.3 Orbit Regularization

We formulate our learning problem as follows:

$$\text{minimize } L(\boldsymbol{w}) \quad \text{s.t.} \quad \boldsymbol{w} \preceq_G \boldsymbol{v}, \tag{3}$$

where $L : V \to \mathbb{R}$ is a loss function, $G$ is a given group, and $\boldsymbol{v} \in V$ is a seed vector. This formulation subsumes several well-known cases, outlined below.

- **$\ell_2$-regularization.** If $G := \mathsf{O}(d)$ is the orthogonal group acting by multiplication, we recover $\ell_2$ regularization. Indeed, we have $G\boldsymbol{v} = \{U\boldsymbol{v} \in \mathbb{R}^d \mid U \in \mathsf{O}(d)\} = \{\boldsymbol{w} \in \mathbb{R}^d \mid \|\boldsymbol{w}\|_2 = \|\boldsymbol{v}\|_2\}$, for any seed $\boldsymbol{v} \in \mathbb{R}^d$. That is, the orbitope $\mathcal{O}_G(\boldsymbol{v}) = \mathbf{conv}(G\boldsymbol{v})$ becomes the $\ell_2$-ball with radius $\|\boldsymbol{v}\|_2$. The only property of the seed that matters in this case is its $\ell_2$-norm.

- **Permutahedron.** Let $\mathcal{P}$ be the *symmetric group* (also called the *permutation group*), which can be represented as the set of $d$-by-$d$ permutation matrices. Given $\boldsymbol{v} \in \mathbb{R}^d$, the orbitope induced by $\boldsymbol{v}$ under $\mathcal{P}$ is the convex hull of all the permutations of $\boldsymbol{v}$, which can be equivalently described as the vectors that are transformations of $\boldsymbol{v}$ through a doubly stochastic matrix:

$$\mathcal{O}_{\mathcal{P}}(\boldsymbol{v}) = \mathbf{conv}\{P\boldsymbol{v} \mid P \in \mathcal{P}\} = \{M\boldsymbol{v} \mid M\mathbf{1} = \mathbf{1},\ M^\top \mathbf{1} = \mathbf{1},\ M \geq 0\}. \tag{4}$$

This set is called the *permutahedron* [12]. We will revisit this case in §4.

- **Signed permutahedron.** Let $\mathcal{P}_\pm$ be the *hyperoctahedral group* (also called the *signed permutation group*), *i.e.*, the $d$-by-$d$ matrices with entries in $\{0, \pm 1\}$ such that the sum of the absolute values in each row and column is 1. The action of $\mathcal{P}_\pm$ on $\mathbb{R}^d$ permutes the entries of vectors and arbitrarily switches signs. Given $\boldsymbol{v} \in \mathbb{R}^d$, the orbitope induced by $\boldsymbol{v}$ under $\mathcal{P}_\pm$ is:

$$\mathcal{O}_{\mathcal{P}_\pm}(\boldsymbol{v}) = \mathbf{conv}\{\mathbf{Diag}(\boldsymbol{s})P\boldsymbol{v} \mid P \in \mathcal{P},\ \boldsymbol{s} \in \{\pm 1\}^d\}, \tag{5}$$

where $\mathbf{Diag}(\boldsymbol{s})$ denotes a diagonal matrix formed by the entries of $\boldsymbol{s}$. We call this set the *signed permutahedron*; it is depicted in Fig. 1 and will also be revisited in §4.

- **$\ell_1$ and $\ell_\infty$-regularization.** As a particular case of the signed permutahedron, we recover $\ell_1$ and $\ell_\infty$ balls by choosing seeds of the form $\boldsymbol{v} = \gamma \boldsymbol{e}_1$ (a scaled canonical basis vector) and $\boldsymbol{v} = \gamma \mathbf{1}$ (a constant vector), respectively, where $\gamma$ is a scalar. In the first case, we obtain the $\ell_1$-ball, $\mathcal{O}_G(\boldsymbol{v}) = \gamma \mathbf{conv}(\{\boldsymbol{e}_1, \ldots, \boldsymbol{e}_d\})$ and in the second case, we get the $\ell_\infty$-ball $\mathcal{O}_G(\boldsymbol{v}) = \gamma \mathbf{conv}(\{\pm 1\}^d)$.

- **Symmetric matrices with majorized eigenvalues.** Let $G := \mathsf{O}(d)$ be again the orthogonal group, but now acting by conjugation on the vector space of $d$-by-$d$ symmetric matrices, $V = \mathbb{S}^d$. Given a seed $\boldsymbol{v} \equiv A \in \mathbb{S}^d$, its orbit is $G\boldsymbol{v} = \{UAU^\top \mid U \in \mathsf{O}(d)\} = \{U\,\mathbf{Diag}(\boldsymbol{\lambda}(A))U^\top \mid U \in \mathsf{O}(d)\}$, where $\boldsymbol{\lambda}(A)$ denotes a vector containing the eigenvalues of $A$ in decreasing order (so we may assume without loss of generality that the seed is diagonal). The orbitope $\mathcal{O}_G(\boldsymbol{v})$ becomes:

$$\mathcal{O}_G(\boldsymbol{v}) := \{B \in \mathbb{S}^d \mid \boldsymbol{\lambda}(B) \preceq_{\mathcal{P}} \boldsymbol{\lambda}(A)\}, \tag{6}$$

which is the set of matrices whose eigenvalues are in the permutahedron $\mathcal{O}_{\mathcal{P}}(\boldsymbol{\lambda}(A))$ (see example above). This is called the *Schur-Horn orbitope* in the literature [8].

- **Squared matrices with majorized singular values.** Let $G := \mathsf{O}(d) \times \mathsf{O}(d)$ act on $\mathbb{R}^{d \times d}$ (the space of squared matrices, not necessarily symmetric) as $g_{U,V}A := UAV^\top$. Given a seed $\boldsymbol{v} \equiv A$, its orbit is $G\boldsymbol{v} = \{UAV^\top \mid U, V \in \mathsf{O}(d)\} = \{U\,\mathbf{Diag}(\boldsymbol{\sigma}(A))V^\top \mid U, V \in \mathsf{O}(d)\}$, where $\boldsymbol{\sigma}(A)$ contains the singular values of $A$ in decreasing order (so we may assume without loss of generality that the seed is diagonal and non-negative). The orbitope $\mathcal{O}_G(\boldsymbol{v})$ becomes:

$$\mathcal{O}_G(\boldsymbol{v}) := \{B \in \mathbb{R}^{d \times d} \mid \boldsymbol{\sigma}(B) \preceq_{\mathcal{P}} \boldsymbol{\sigma}(A)\}, \tag{7}$$

which is the set of matrices whose singular values are in the permutahedron $\mathcal{O}_{\mathcal{P}}(\boldsymbol{\sigma}(A))$.

- **Spectral and nuclear norm regularization.** The previous case subsumes spectral and nuclear norm balls: indeed, for a seed $A = \gamma I_d$, the orbitope becomes the convex hull of orthogonal matrices, which is the spectral norm ball $\{A \in \mathbb{R}^{d \times d} \mid \|A\|_2 := \sigma_1(A) \leq \gamma\}$; while for a seed $A = \gamma\,\mathbf{Diag}(\boldsymbol{e}_1)$, the orbitope becomes the convex hull of rank-1 matrices with norm bounded by $\gamma$, which is the nuclear norm ball $\{A \in \mathbb{R}^{d \times d} \mid \|A\|_* := \sum_i \sigma_i \leq \gamma\}$. This norm has been widely used for low-rank matrix factorization and matrix completion [3].

Besides these examples, other regularization strategies, such as non-overlapping $\ell_{2,1}$ and $\ell_{\infty,1}$ norms [13, 4] can be obtained by considering products of the groups above. We omit details for space.

## 2.4 Relation with Atomic Norms

Atomic norms have been recently proposed as a toolbox for structured sparsity [6]. Let $\mathcal{A} \subseteq V$ be a centrally symmetric set of atoms, *i.e.*, $\boldsymbol{v} \in \mathcal{A}$ iff $-\boldsymbol{v} \in \mathcal{A}$. The atomic norm induced by $\mathcal{A}$ is defined as $\|\boldsymbol{w}\|_{\mathcal{A}} := \inf\{t > 0 \mid \boldsymbol{w} \in t\,\mathbf{conv}(\mathcal{A})\}$. The corresponding atomic ball is the set $\{\boldsymbol{w} \mid \|\boldsymbol{w}\|_{\mathcal{A}} \leq 1\} = \mathbf{conv}(\mathcal{A})$. Not surprisingly, orbitopes are often atomic norm balls.

**Proposition 3 (Atomic norms)** *If $G$ is a subgroup of the* general linear group $\mathsf{GL}(d)$ *and satisfies* $-\boldsymbol{v} \in G\boldsymbol{v}$, *then the set $\mathcal{O}_G(\boldsymbol{v})$ is the ball of an atomic norm.*

*Proof:* Under the given assumption, the set $G\boldsymbol{v}$ is centrally symmetric, *i.e.*, it satisfies $\boldsymbol{w} \in G\boldsymbol{v}$ iff $-\boldsymbol{w} \in G\boldsymbol{v}$ (indeed, the left hand side implies that $\boldsymbol{w} = g\boldsymbol{v}$ for some $g \in G$, and $-\boldsymbol{v} \in G\boldsymbol{v}$ implies that $-\boldsymbol{v} = h\boldsymbol{v}$ for some $h \in G$, therefore, $-\boldsymbol{w} = -gh^{-1}(-\boldsymbol{v}) = gh^{-1}\boldsymbol{v} \in G\boldsymbol{v}$). As shown by Chandrasekaran et al. [6], this guarantees that $\|.\|_{G\boldsymbol{v}}$ satisfies the axioms of a norm. ∎

**Corollary 4** *For any choice of seed, the signed permutahedron $\mathcal{O}_{\mathcal{P}_\pm}(\boldsymbol{v})$ and the orbitope formed by the squared matrices with majorized singular values are both atomic norm balls. If $d$ is even and $\boldsymbol{v}$ is of the form $\boldsymbol{v} = (\boldsymbol{v}_+, -\boldsymbol{v}_+)$, with $\boldsymbol{v}_+ \in \mathbb{R}_+^{d/2}$, then the permutahedron $\mathcal{O}_{\mathcal{P}}(\boldsymbol{v})$ and the orbitope formed by the symmetric matrices with eigenvalues majorized by $\boldsymbol{\lambda}(\boldsymbol{v})$ are both atomic norm balls.*

## 3 Matching Function and Region Cones

We now construct a unifying perspective that highlights the role of the group $G$. Two key concepts that play a crucial role in our analysis are that of *matching function* and *region cone*. In the sequel, these will work as building blocks for important algorithmic and geometric characterizations.

**Definition 5 (Matching function)** *The matching function of $G$, $m_G : V \times V \to \mathbb{R}$, is defined as:*

$$m_G(\boldsymbol{u}, \boldsymbol{v}) := \sup\{\langle \boldsymbol{u}, \boldsymbol{w} \rangle \mid \boldsymbol{w} \in G\boldsymbol{v}\}. \tag{8}$$

Intuitively, $m_G(\boldsymbol{u}, \boldsymbol{v})$ "aligns" the orbits of $\boldsymbol{u}$ and $\boldsymbol{v}$ before taking the inner product. Note also that $m_G(\boldsymbol{u}, \boldsymbol{v}) = \sup\{\langle \boldsymbol{u}, \boldsymbol{w} \rangle \mid \boldsymbol{w} \in \mathcal{O}_G(\boldsymbol{v})\}$, since we may equivalently maximize the linear objective over $\mathcal{O}_G(\boldsymbol{v})$, which is the convex hull of $G\boldsymbol{v}$. We therefore have the following

**Proposition 6 (Duality)** *Fix $\boldsymbol{v} \in V$, and define the indicator function of the orbitope, $I_{\mathcal{O}_G(\boldsymbol{v})}(\boldsymbol{w}) = 0$ if $\boldsymbol{w} \in \mathcal{O}_G(\boldsymbol{v})$, and $-\infty$ otherwise. The Fenchel dual of $I_{\mathcal{O}_G(\boldsymbol{v})}$ is $m_G(., \boldsymbol{v})$. As a consequence, letting $L^\star : V \to \mathbb{R}$ is the Fenchel dual of the loss $L$, the dual problem of Eq. 3 is:*

$$\text{maximize } - L^\star(-\boldsymbol{u}) - m_G(\boldsymbol{u}, \boldsymbol{v}) \quad \text{w.r.t.} \quad \boldsymbol{u} \in V. \tag{9}$$

Note that if $\|.\|_{G\boldsymbol{v}}$ is a norm (*e.g.*, if the conditions of Prop. 3 are satisfied), then the statement above means that $m_G(., \boldsymbol{v}) = \|.\|_{G\boldsymbol{v}}^\star$ is its dual norm. We will revisit this dual formulation in §4.

The following properties have been established in [14, 15].

**Proposition 7** *For any $\boldsymbol{u}, \boldsymbol{v} \in V$, we have: (i) $m_G(c_1\boldsymbol{u}, c_2\boldsymbol{v}) = c_1 c_2 m_G(\boldsymbol{u}, \boldsymbol{v})$ for $c_1, c_2 \geq 0$; (ii) $m_G(g_1\boldsymbol{u}, g_2\boldsymbol{v}) = m_G(\boldsymbol{u}, \boldsymbol{v})$ for $g_1, g_2 \in G$; (iii) $m_G(\boldsymbol{u}, \boldsymbol{v}) = m_G(\boldsymbol{v}, \boldsymbol{u})$. Furthermore, the following three statements are equivalent: (i) $\boldsymbol{w} \preceq_G \boldsymbol{v}$, (ii) $f(\boldsymbol{w}) \leq f(\boldsymbol{v})$ for all $G$-invariant convex functions $f : V \to \mathbb{R}$, (iii) $m_G(\boldsymbol{u}, \boldsymbol{w}) \leq m_G(\boldsymbol{u}, \boldsymbol{v})$ for all $\boldsymbol{u} \in V$.*

In the sequel, we always assume that $G$ is a subgroup of the orthogonal group $\mathsf{O}(d)$. This implies that the orbitope $\mathcal{O}_G(\boldsymbol{v})$ is compact for any $\boldsymbol{v} \in V$ (and therefore the sup in Eq. 8 can be replaced by a max), and that $\|g\boldsymbol{v}\| = \|\boldsymbol{v}\|$ for any $\boldsymbol{v} \in V$. Another important concept is that of the *normal cone* of a point $\boldsymbol{w} \in V$ with respect to the orbitope $\mathcal{O}_G(\boldsymbol{v})$, denoted as $N_{G\boldsymbol{v}}(\boldsymbol{w})$ and defined as follows:

$$N_{G\boldsymbol{v}}(\boldsymbol{w}) := \{\boldsymbol{u} \in V \mid \langle \boldsymbol{u}, \boldsymbol{w}' - \boldsymbol{w} \rangle \leq 0 \; \forall \boldsymbol{w}' \preceq_G \boldsymbol{v}\}. \tag{10}$$

Normal cones plays an important role in convex analysis [16]. The particular case of the normal cone at the seed $\boldsymbol{v}$ (illustrated in Fig. 1) is of great importance, as will be seen below.

**Definition 8 (Region cone)** *Given $\boldsymbol{v} \in V$, the region cone at $\boldsymbol{v}$ is $K_G(\boldsymbol{v}) := N_{G\boldsymbol{v}}(\boldsymbol{v})$. It is the set of points that are "maximally aligned" with $\boldsymbol{v}$ in terms of the matching function:*

$$K_G(\boldsymbol{v}) \;\; = \;\; \{\boldsymbol{u} \in V \mid m_G(\boldsymbol{u}, \boldsymbol{v}) = \langle \boldsymbol{u}, \boldsymbol{v} \rangle\}. \tag{11}$$

## 4 Permutahedra and Sorted $\ell_1$-Norms

In this section, we focus on the permutahedra introduced in §2. Below, given a vector $\boldsymbol{w} \in \mathbb{R}^d$, we denote by $w_{(k)}$ its $k$th order statistic, *i.e.*, we will "sort" $\boldsymbol{w}$ so that $w_{(1)} \geq w_{(2)} \geq \ldots \geq w_{(d)}$. We also consider the order statistics of the magnitudes $|w|_{(k)}$ by sorting the absolute values.

## 4.1 Signed Permutahedron

We start by defining the "sorted $\ell_1$-norm," proposed by Bogdan et al. [9] in their recent SLOPE method as a means to control the false discovery rate, and studied by Zeng and Figueiredo [17].

**Definition 9 (Sorted $\ell_1$-norm)** *Let $v, w \in \mathbb{R}^d$, with $v_1 \geq v_2 \geq \ldots \geq v_d \geq 0$ and $v_1 > 0$. The sorted $\ell_1$-norm of $w$ (weighted by $v$) is defined as: $\|w\|_{\text{SLOPE},v} := \sum_{j=1}^d v_j |w|_{(j)}$.*

In [9] it is shown that $\|.\|_{\text{SLOPE},v}$ satisfies the axioms of a norm. The rationale is that larger components of $w$ are penalized more than smaller ones, in a way controlled by the prescribed $v$. For $v = 1$, we recover the standard $\ell_1$-norm, while the $\ell_\infty$-norm corresponds to $v = e_1$. Another special case is the OSCAR regularizer [18, 19], $\|w\|_{\text{OSCAR},\tau_1,\tau_2} := \tau_1 \|w\|_1 + \tau_2 \sum_{i<j} \max\{|w_i|, |w_j|\}$, corresponding to a linearly spaced $v$, $v_j = (\tau_1 + \tau_2(d - j))$ for $j = 1, \ldots, d$. The next proposition reveals a connection between SLOPE and the atomic norm induced by the signed permutahedron.

**Proposition 10** *Let $v \in \mathbb{R}_+^d$ be as in Def. 9. The sorted $\ell_1$-norm weighted by $v$ and the atomic norm induced by the $\mathcal{P}^\pm$-orbitope seeded at $v$ are dual to each other: $\|.\|_{\mathcal{P}_\pm v}^\star = \|.\|_{\text{SLOPE},v}$.*

*Proof:* From Prop. 6, we have $\|w\|_{\mathcal{P}_\pm v}^\star = m_{\mathcal{P}_\pm}(w, v)$. Let $P$ be a signed permutation matrix s.t. $\tilde{w} := Pw$ has its components sorted by decreasing magnitude, $|\tilde{w}|_1 \geq \ldots \geq |\tilde{w}|_d$. From Prop. 7, we have $m_{\mathcal{P}_\pm}(w, v) = m(\tilde{w}, v) = \langle |\tilde{w}|, v \rangle = \|w\|_{\text{SLOPE},v}$. ∎

The next proposition [7, 14] provides a characterization of the $\mathcal{P}_\pm$-orbitope in terms of inequalities about the cumulative distribution of the order statistics.

**Proposition 11 (Submajorization ordering)** *The orbitope $\mathcal{O}_{\mathcal{P}_\pm}(v)$ can be characterized as:*

$$\mathcal{O}_{\mathcal{P}_\pm}(v) = \left\{ w \in \mathbb{R}^d \mid \sum_{j \leq i} |w|_{(j)} \leq \sum_{j \leq i} |v|_{(j)}, \ \forall i = 1, \ldots, d \right\}. \tag{12}$$

Prop. 11 leads to a precise characterization of the atomic norm $\|w\|_{\mathcal{P}_\pm v}$, and therefore of the dual norm of SLOPE: $\|w\|_{\mathcal{P}_\pm v} = \max_{i=1,\ldots,d} \sum_{j \leq i} |w|_{(j)} / \sum_{j \leq i} |v|_{(j)}$.

## 4.2 Permutahedron

The unsigned counterpart of Prop. 11 goes back to Hardy et al. [20].

**Proposition 12 (Majorization ordering)** *The $\mathcal{P}$-orbitope seeded at $v$ can be characterized as:*

$$\mathcal{O}_{\mathcal{P}}(v) = \left\{ w \in \mathbb{R}^d \mid 1^\top w = 1^\top v \ \wedge \ \sum_{j \leq i} w_{(j)} \leq \sum_{j \leq i} v_{(j)}, \ \forall i = 1, \ldots, d - 1 \right\}. \tag{13}$$

As seen in Corollary 4, if $d$ is even and $v = (v_+, -v_+)$, with $v \geq 0$, then $\|w\|_{\mathcal{P}v}$ qualifies as a norm (we need to confine to the linear subspace $V := \{w \in \mathbb{R}^d \mid \sum_{j=1}^d w_j = 0\}$). From Prop. 12, we have that this norm can be written as: $\|w\|_{\mathcal{P}v} = \max_{i=1,\ldots,d-1} \sum_{j \leq i} w_{(j)} / \sum_{j \leq i} v_{(j)}$.

**Proposition 13** *Assume the conditions above hold and that $v_1 \geq v_2 \geq \ldots \geq v_{d/2} \geq 0$ and $v_1 > 0$. The dual norm of $\|.\|_{\mathcal{P}v}$ is $\|w\|_{\mathcal{P}v}^\star = \sum_{j=1}^{d/2} v_j (w_{(j)} - w_{(d-j+1)})$.*

*Proof:* Similar to the proof of Prop. 11. ∎

## 5 Conditional and Projected Gradient Algorithms

Two important classes of algorithms in sparse modeling are the *conditional gradient method* [21, 22] and the *proximal gradient method* [23, 24]. Under Ivanov regularization as in Eq. 3, the latter reduces to the projected gradient method. In this section, we show that both algorithms are a good fit for solving Eq. 3 for arbitrary groups, as long as the two building blocks mentioned in §3 are available: *(i)* a procedure for evaluating the matching function (necessary for conditional gradient methods) and *(ii)* a procedure for projecting onto the region cone (necessary for projected gradient).

```
1: Initialize $\boldsymbol{w}_1 = \mathbf{0}$                              1: Initialize $\boldsymbol{w}_1 = \mathbf{0}$
2: for $t = 1, 2, \ldots$ do                                              2: for $t = 1, 2, \ldots$ do
3:     $\boldsymbol{u}_t = \arg\max_{\boldsymbol{u} \preceq_G \boldsymbol{v}} \langle -\nabla L(\boldsymbol{w}_t), \boldsymbol{u} \rangle$    3:     Choose a stepsize $\eta_t$
4:     $\eta_t = 2/(t+2)$                                                 4:     $\boldsymbol{a} = \boldsymbol{w}_t - \eta_t \nabla L(\boldsymbol{w}_t)$
5:     $\boldsymbol{w}_{t+1} = (1 - \eta_t)\boldsymbol{w}_t + \eta_t \boldsymbol{u}_t$     5:     $\boldsymbol{w}_{t+1} = \arg\min_{\boldsymbol{w} \preceq_G \boldsymbol{v}} \|\boldsymbol{w} - \boldsymbol{a}\|$
6: end for                                                                6: end for
```

**Figure 2:** Conditional gradient (left) and projected gradient (right) algorithms.

### 5.1 Conditional Gradient

The conditional gradient method is shown in Fig. 2 (left). We assume that a procedure is available for computing the gradient of the loss. The relevant part is the maximization in line 3, which corresponds precisely to an evaluation of the matching function $m(\boldsymbol{s}, \boldsymbol{v})$, with $\boldsymbol{s} = -\nabla L(\boldsymbol{w}_t)$ (cf. Eq. 8). Fortunately, this step is efficient for a variety of cases:

**Permutations.** If $G = \mathcal{P}$, the matching function can be evaluated in time $O(d \log d)$ with a simple sort operation. Without losing generality, we assume the seed $\boldsymbol{v}$ is sorted in descending order (otherwise, pre-sort it before the main loop starts). Then, each time we need to evaluate $m(\boldsymbol{s}, \boldsymbol{v})$, we compute a permutation $P$ such that $P\boldsymbol{s}$ is also sorted. The minimizer in line 3 will equal $P^{-1}\boldsymbol{v}$.

**Signed permutations.** If $G = \mathcal{P}_{\pm}$, a similar procedure with the same $O(d \log d)$ runtime also works, except that now we sort the absolute values, and set the signs of $P^{-1}\boldsymbol{v}$ to match those of $\boldsymbol{s}$.

**Symmetric matrices with majorized eigenvalues.** Let $A = U_A \boldsymbol{\lambda}(A) U_A^\top \in \mathbb{S}^d$ and $B = U_B \boldsymbol{\lambda}(B) U_B^\top \in \mathbb{S}^d$, where the eigenvalues $\boldsymbol{\lambda}(A)$ and $\boldsymbol{\lambda}(B)$ are sorted in decreasing order. In this case, the matching function becomes $m_G(A, B) = \max_{V \in \mathsf{O}(d)} \mathbf{trace}(A^\top V B V^\top) = \langle \boldsymbol{\lambda}(A), \boldsymbol{\lambda}(B) \rangle$ due to von Neumann's trace inequality [25], the maximizer being $V = U_A U_B^\top$. Therefore, we need only to make an eigen-decomposition and set $B' = U_A \boldsymbol{\lambda}(B) U_A^\top$.

**Squared matrices with majorized singular values.** Let $A = U_A \boldsymbol{\sigma}(A) V_A^\top \in \mathbb{R}^{d \times d}$ and $B = U_B \boldsymbol{\sigma}(B) V_B^\top \in \mathbb{R}^{d \times d}$, where the singular values are sorted. We have $m_G(A, B) = \max_{U, V \in \mathsf{O}(d)} \mathbf{trace}(A^\top U B V^\top) = \langle \boldsymbol{\sigma}(A), \boldsymbol{\sigma}(B) \rangle$ also from von Neumann's inequality [25]. To evaluate the matching function, we need only to make an SVD and set $B' = U_A \boldsymbol{\sigma}(B) V_A^\top$.

### 5.2 Projected Gradient

The projected gradient algorithm is illustrated in Fig. 2 (right); the relevant part is line 5, which involves a projection onto the orbitope $\mathcal{O}_G(\boldsymbol{v})$. This projection may be hard to compute directly, since the orbitope may lack a concise half-space representation. However, we next transform this problem into a projection onto the region cone $K_G(\boldsymbol{v})$ (the proof is in the supplemental material).

**Proposition 14** *Assume $G$ is a subgroup of $\mathsf{O}(d)$. Let $g \in G$ be such that $\langle \boldsymbol{a}, g\boldsymbol{v} \rangle = m_G(\boldsymbol{a}, \boldsymbol{v})$. Then, the solution of the problem in line 5 is $\boldsymbol{w}^* = \boldsymbol{a} - \Pi_{K_G(g\boldsymbol{v})}(\boldsymbol{a} - g\boldsymbol{v})$.*

Thus, all is necessary is computing the arg-max associated with the matching function, and a black box that projects onto the region cone $K_G(\boldsymbol{v})$. Again, this step is efficient in several cases:

**Permutations.** If $G = \mathcal{P}$, the region cone of a point $\boldsymbol{v}$ is the set of points $\boldsymbol{w}$ satisfying $v_i > v_j \Rightarrow w_i \geq w_j$, for all $i, j \in 1, \ldots, d$. Projecting onto this cone is a well-studied problem in isotonic regression [26, 27], with existing $O(d)$ algorithms.

**Signed permutations.** If $G = \mathcal{P}_{\pm}$, this problem is precisely the evaluation of the proximity operator of the sorted $\ell_1$-norm, also solvable in $O(d)$ runtime with a stack-based algorithm [9].

## 6 Continuation Algorithm

Finally, we present a general continuation procedure for exploring regularization paths when $L$ is a convex loss function (not necessarily differentiable) and the seed $\boldsymbol{v}$ is not prescribed. The

**Require:** Factor $\epsilon > 0$, interpolation parameter $\alpha \in [0, 1]$
1: Initialize seed $\boldsymbol{v}_0$ randomly and set $\|\boldsymbol{v}_0\| = \epsilon$
2: Set $t = 0$
3: **repeat**
4:    Solve $\boldsymbol{w}_t = \arg\min_{\boldsymbol{w} \preceq_G \boldsymbol{v}_t} L(\boldsymbol{w})$
5:    Pick $\boldsymbol{v}'_t \in G\boldsymbol{v}_t \cap K_G(\boldsymbol{w}_t)$
6:    Set next seed $\boldsymbol{v}_{t+1} = (1 + \epsilon)(\alpha \boldsymbol{v}'_t + (1 - \alpha)\boldsymbol{w}_t)$
7:    $t \leftarrow t + 1$
8: **until** $\|\boldsymbol{w}_t\|_{G\boldsymbol{v}_t} < 1$.
9: Use cross-validation to choose the best $\widehat{\boldsymbol{w}} \in \{\boldsymbol{w}_1, \boldsymbol{w}_2, \dots\}$

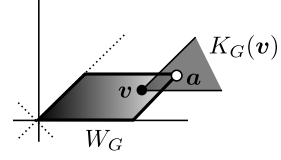

**Figure 3:** Left: Continuation algorithm. Right: Reachable region $W_G$ for the hyperoctahedral group, with a reconstruction loss $L(\boldsymbol{w}) = \|\boldsymbol{w} - \boldsymbol{a}\|^2$. Only points $\boldsymbol{v}$ s.t. $-\nabla L(\boldsymbol{v}) = \boldsymbol{a} - \boldsymbol{v} \in K_G(\boldsymbol{v})$ belong to this set. Different initializations of $\boldsymbol{v}_0$ lead to different paths along $W_G$, all ending in $\boldsymbol{a}$.

procedure—outlined in Fig. 3—solves instances of Eq. 3 for a sequence of seeds $\boldsymbol{v}_1, \boldsymbol{v}_2, \dots$, using a simple heuristic for choosing the next seed given the previous one and the current solution.

The basic principle behind this procedure is the same as in other homotopy continuation methods [28, 29, 30, 31]: we start with very strong regularization (using a small norm ball), and then gradually weaken the regularization (increasing the ball) while "tracking" the solution. The process stops when the solution is found to be in the interior of the ball (the condition in line 8), which means the regularization constraint is no longer active. The main difference with respect to classical homotopy methods is that we do not just *scale* the ball (in our case, the $G$-orbitope); we also generate new seeds that *shape* the ball along the way. To do so, we adopt a simple heuristic (line 6) to make the seed move toward the current solution $\boldsymbol{w}_t$ before scaling the orbitope. This procedure depends on the initialization (see Fig. 3 for an illustration), which drives the search into different regions. Reasoning in terms of groups, line 4 makes us move *inside* the orbits, while line 6 is an heuristic to *jump* to a nearby orbit. For any choice of $\epsilon > 0$ and $\alpha \in [0, 1]$, the algorithm is convergent and produces a strictly decreasing sequence $L(\boldsymbol{w}_1) > L(\boldsymbol{w}_2) > \cdots$ before it terminates (a proof is provided as supplementary material). We expect that, eventually, a seed $\boldsymbol{v}$ will be generated that is close to the true model $\widehat{\boldsymbol{w}}$. Although it may not be obvious at first sight why would it be desirable that $\boldsymbol{v} \approx \widehat{\boldsymbol{w}}$, we provide a simple result below (Prop. 15) that sheds some light on this matter, by characterizing the set of points in $V$ that are "reachable" by optimizing Eq. 3.

From the optimality conditions of convex programming [32, p. 257], we have that $\boldsymbol{w}^*$ is a solution of the optimization problem in Eq. 3 if and only if $\mathbf{0} \in \partial L(\boldsymbol{w}^*) + N_{G\boldsymbol{v}}(\boldsymbol{w}^*)$, where $\partial L(\boldsymbol{w})$ denotes the subdifferential of $L$ at $\boldsymbol{w}$, and $N_{G\boldsymbol{v}}(\boldsymbol{w})$ is the normal cone to $\mathcal{O}_G(\boldsymbol{v})$ at $\boldsymbol{w}$, defined in §3. For certain seeds $\boldsymbol{v} \in V$, it may happen that the optimal solution $\boldsymbol{w}^*$ of Eq. 3 is the seed itself. Let $W_G$ be the set of seeds with this property:

$$W_G := \{\boldsymbol{v} \in V \mid L(\boldsymbol{v}) \leq L(\boldsymbol{w}), \ \forall \boldsymbol{w} \preceq_G \boldsymbol{v}\} = \{\boldsymbol{v} \in V \mid \mathbf{0} \in \partial L(\boldsymbol{v}) + K_G(\boldsymbol{v})\}, \quad (14)$$

where $K_G(\boldsymbol{v})$ is the region cone and the right hand side follows from the optimality conditions. We next show that this set is all we need to care about.

**Proposition 15** *Consider the set of points that are solutions of Eq. 3 for* some *seed* $\boldsymbol{v} \in V$, $\widehat{W}_G :=$ $\left\{\boldsymbol{w}^* \in V \mid \exists \boldsymbol{v} \in V \ : \ \boldsymbol{w}^* \in \arg\min_{\boldsymbol{w} \preceq_G \boldsymbol{v}} L(\boldsymbol{w})\right\}$. *We have* $\widehat{W}_G = W_G$.

*Proof:* Obviously, $\boldsymbol{v} \in W_G \Rightarrow \boldsymbol{v} \in \widehat{W}_G$. For the reverse direction, suppose that $\boldsymbol{w}^* \in \widehat{W}_G$, in which case there is some $\boldsymbol{v} \in V$ such that $\boldsymbol{w}^* \preceq_G \boldsymbol{v}$ and $L(\boldsymbol{w}^*) \leq L(\boldsymbol{w})$ for any $\boldsymbol{w} \preceq_G \boldsymbol{v}$. Since $\preceq_G$ is a pre-order, it must hold in particular that $L(\boldsymbol{w}^*) \leq L(\boldsymbol{w})$ for any $\boldsymbol{w} \preceq_G \boldsymbol{w}^* \preceq_G \boldsymbol{v}$. Therefore, we also have that $\boldsymbol{w}^* \in \arg\min_{\boldsymbol{w} \preceq_G \boldsymbol{w}^*} L(\boldsymbol{w})$, *i.e.*, $\boldsymbol{w}^* \in W_G$. ∎

## 7  Simulation Results

We describe the results of numerical experiments when regularizing with the permutahedron (symmetric group) and the signed permutahedron (hyperoctahedral group). All problems were solved using the conditional gradient algorithm, as described in §5. We generated the true model $\widehat{\boldsymbol{w}} \in \mathbb{R}^d$

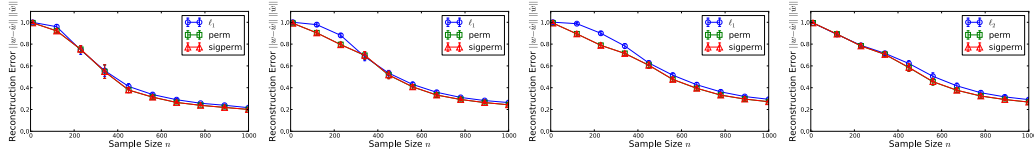

**Figure 4:** Learning curves for the permutahedron and signed permutahedron regularizers with a perfect seed. Shown are averages and standard deviations over 10 trials. The baselines are $\ell_1$ (three leftmost plots, resp. with $k = 150, 250, 400$), and $\ell_2$ (last plot, with $k = 500$).

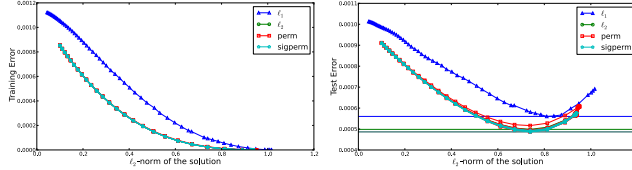

**Figure 5:** Mean squared errors in the training set (left) and the test set (right) along the regularization path. For the permutahedra regularizers, this path was traced with the continuation algorithm. The baseline is $\ell_1$ regularization. The horizontal lines in the right plot show the solutions found with validation in a held-out set.

by sampling the entries from a uniform distribution in $[0, 1]$ and subtracted the mean, keeping $k \leq d$ nonzeros; after which $\widehat{w}$ was normalized to have unit $\ell_2$-norm. Then, we sampled a random $n$-by-$d$ matrix $X$ with i.i.d. Gaussian entries and variance $\sigma^2 = 1/d$, and simulated measurements $y = X\widehat{w} + n$, where $n \sim N(0, \sigma_n^2)$ is Gaussian noise. We set $d = 500$ and $\sigma_n = 0.3\sigma$.

For the first set of experiments (Fig. 4), we set $k \in \{150, 250, 400, 500\}$ and varied the number of measurements $n$. To assess the advantage of knowing the true parameters up to a group transformation, we used for the orbitope regularizers a seed in the orbit of the true $\widehat{w}$, up to a constant factor (this constant, and the regularization constants for $\ell_1$ and $\ell_2$, were all chosen with validation in a held-out set). As expected, this information was beneficial, and no significant difference was observed between the permutahedron and the signed permutahedron. For the second set of experiments (Fig. 5), where the aim is to assess the performance of the continuation method, no information about the true model was given. Here, we fixed $n = 250$ and $k = 300$ and ran the continuation algorithm with $\epsilon = 0.1$ and $\alpha = 0.0$, for 5 different initializations of $v_0$. We observe that this procedure was effective at exploring the orbits, eventually finding a slightly better model than the one found with $\ell_1$ and $\ell_2$ regularizers.

# 8 Conclusions and Future Work

In this paper, we proposed a group-based regularization scheme using the notion of orbitopes. Simple choices of groups recover commonly used regularizers such as $\ell_1$, $\ell_2$, $\ell_\infty$, spectral and nuclear matrix norms; as well as some new ones, such as the permutahedron and signed permutahedron. As a byproduct, we revealed a connection between the permutahedra and the recently proposed sorted $\ell_1$-norm. We derived procedures for learning with these orbit regularizers via conditional and projected gradient algorithms, and a continuation strategy for orbit exploration.

There are several avenues for future research. For example, certain classes of groups, such as reflection groups [33], have additional properties that may be exploited algorithmically. Our work should be regarded as a first step toward group-based regularization—we believe that the regularizers studied here are just the tip of the iceberg. Groups and their representations are well studied in other disciplines [10], and chances are high that this framework can lead to new regularizers that are a good fit to specific machine learning problems.

**Acknowledgments**

We thank all reviewers for their valuable comments. This work was partially supported by FCT grants PTDC/EEI-SII/2312/2012 and PEst-OE/EEI/LA0008/2011, and by the EU/FEDER programme, QREN/POR Lisboa (Portugal), under the Intelligo project (contract 2012/24803).

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
