[Supplementary Material]

# Supplementary Material

## A  Proof of Prop. 2

Reflexivity ($v \preceq_G v, \forall v \in V$) comes from the fact that the identity element $1_G$ leaves any $v$ unchanged, and therefore $v \in Gv \subseteq \mathcal{O}_G(v)$. To prove transitivity, it suffices to show that

$$\mathcal{O}_G(w) \subseteq \mathcal{O}_G(v) \quad \Leftrightarrow \quad w \preceq_G v. \tag{15}$$

The direct statement ($\Rightarrow$) follows from reflexivity: since $w \in \mathcal{O}_G(w) \subseteq \mathcal{O}_G(v)$, this implies $w \in \mathcal{O}_G(v)$. For the converse statement ($\Leftarrow$), note that $w \in \mathcal{O}_G(v)$ implies that $w = \sum_i c_i h_i v$ for $\{h_i\} \subseteq G$ and non-negative scalars $\{c_i\}$ that sum to one; using the linearity of the action, we then have that $gw = \sum_i c_i g h_i v \in \mathcal{O}_G(v)$ for any $g \in G$, which implies $Gw \subseteq \mathcal{O}_G(v)$ and $\mathcal{O}_G(w) \subseteq \mathcal{O}_G(v)$ (due to convexity of $\mathcal{O}_G(v)$).

## B  Proof of Prop. 14

Let us start by noting that, for arbitrary $h \in G$,

$$
\begin{aligned}
\min_{h \in G} \frac{1}{2}\|hw - a\|^2 &= \min_{h \in G} \frac{1}{2}\|hw\|^2 - \langle hw, a \rangle + \frac{1}{2}\|a\|^2 \\
&= \frac{1}{2}\|w\|^2 + \frac{1}{2}\|a\|^2 - m(w, a) \\
&= \frac{1}{2}\|w - \tilde{a}\|^2, 
\end{aligned}
\tag{16}
$$

where $\tilde{a} \in Ga$ is such that $m(w, a) = \langle w, \tilde{a} \rangle$; the optimal $h$ satisfies $\tilde{a} = h^{-1}a$; and the second step is justified by the fact that $G$ is a subgroup of $\mathsf{O}(d)$, hence its action is norm-preserving. Due to Moreau's decomposition theorem [34], we have that the projection in line 5 can be computed via proximal operator associated with $I^\star_{\mathcal{O}_G(v)} = m_G(., v)$; namely we have that the (unique) minimizer $w^*$ in line 5 satisfies $w^* = a - \text{prox}_{m_G(.,v)}(a)$. Evaluating the proximal operator boils down to solving the following problem:

$$
\begin{aligned}
\min_{u \in V} \frac{1}{2}\|u - a\|^2 + m_G(u, v) &= \min_{u \in K_G(a)} \frac{1}{2}\|u - a\|^2 + m_G(u, gv) \\
&= \min_{u \in K_G(gv)} \frac{1}{2}\|u - a\|^2 + \langle u, gv \rangle \\
&= \min_{u \in K_G(gv)} \frac{1}{2}\|u - (a - gv)\|^2 + \text{constant}, 
\end{aligned}
\tag{17}
$$

where we used Eq. 16 and the fact that $gv \in K_G(a)$. This leads to the result.

## C  Proof of Convergence of the Continuation Algorithm

We show that for any $\epsilon > 0$, the sequence $(L(w_1), L(w_2), \ldots)$ is strictly decreasing. Convergence follows from the fact that this sequence is lower bounded by the unregularized objective value $\min_w L(w)$, assumed finite. The proof consists of two steps:

1. Showing that, for any $\epsilon > 0$, $w_t$ lies in the interior of $\mathcal{O}_G(v_{t+1})$. This follows from the fact that $v'_t$, $w_t$, and their convex combination all belong to the region cone $K_G(w_t)$; in this region the pre-order induced by $G$ is a cone ordering w.r.t. the polar cone of $K_G$, from which we can derive $w_t \in O_G(\alpha v'_t + (1 - \alpha)w_t)$, leading to the desired statement.

2. Showing that $(L(w_1), L(w_2), \ldots)$ strictly decreases before the algorithm terminates. This is a simple consequence of the previous fact. Since $w_t \in O_G(v_{t+1})$, we must have $L(w_{t+1}) \leq L(w_t)$. If this holds with equality, then $w_{t+1} = w_t$ is an optimal solution at the $(t+1)$th iteration, but since it lies in the interior of $O_G(v_{t+1})$, we have $\|w_{t+1}\|_{Gv_{t+1}} < 1$ and the algorithm will terminate. Therefore we must have $L(w_{t+1}) < L(w_t)$ for the algorithm to proceed.