[Reviews · NeurIPS 2014]

Submitted by Assigned_Reviewer_9

Summary: The authors re-explain regularization in optimization problems as a constraint of the type "the parameters ${\bf w}$ must belong to the convex set $O$" where the convex set "O" is obtained as the convex hull of all the points of the form $g.v$ where $v$ is some fix vector, $g$ an element from a group and $.$ is a (linear) group action of element $g$ on vector $v$.

More concretely, their main contributions are as follows. (A) they explain how several regularizations can be obtained from their framework. For example, the ball associated to the L1 norm can be explained as the convex hull of the points obtained by flipping the sign and permuting the components of the vector $(1,0,0,..,0)$; (B) they show that given a seed $v$ and a group action associated to a group $G$, the notion of "$w$ is a member of the convex set $O_G(v)$" can be seen as "$v$ is smaller than $w$" under a pre-order; (C) they show that if $-v$ belongs to convex set $O$ then $O$ can be seen as the ball of an atomic norm (as defined in Chandra et al.); (D) they show that the L1-sorted norm equals the dual of the norm associated to the signed-pertumation orbitope; (E) they show how to reinterpret the main steps of conditional and projected gradient algorithms in the language of orbitopes and give a procedure to compute projections onto orbitopes. (F) they provide an heuristic algorithm that iteratively morphs the shape of the ball-norm associated to the regularizer, generalizing the idea of regularization paths.

Quality: There are no technical mistakes in the paper. The idea of morphing the shape of the regularizer's ball-norm is the most interesting idea in my opinion. In this regard, it would be good if the authors could clarify the following. Homotopy methods build complete regularization paths that, after being computed, are used in combination with, for example, cross-validation to find the right amount of regularization to perform. I do not understand why the continuation algorithm stops "at (the) point regularization is not having any effect". It would also appreciate if the authors could say a few words about how the continuation algorithm would performs when $\epsilon = 0$. In other words, the shape of the ball-norm is changed by its size is kept constant. Does the algorithm converge ?

Proposition 10 has a trivial pictorial explanation that might be good to include for the sake of clarity. In particular, taking the dual of the norm associated to singed-permutations corresponds to transforming the edges in the ball-norm of Fig. 1-right to vertices and transforming vertices to edges. This leads immediately to the ball-norm of the sorted L1-norm, that can be seen as the intersection of the ball-norms of all weighted L1-norms obtained by permuting the coefficients $w$.

In Prop. 3 the authors show that (under some conditions) orbit regularizers can be seen as atomic norms. It would be good to explain when/how atomic norms can be seen as orbit regularizers.

Clarity: The paper is overall very well written and clear. Here are a few minor things that can be improved. In Line 071 there is a parenthesis missing. In line 244-246 subscripts are missing in $m({\bf w},{\bf v})$, having them would be better. The quality of the pictures should be improved. Are pictures vector format? When I print the paper they look blurred. It would be very useful to have numbers in the references, [1], [2], etc. In Fig. 4, what is the scale in the y-axis referring too?

Significance: I find the idea of morphing the shape of the regularizer's ball-norm potentially interesting. Unfortunately, the numerical results do not clearly show that the continuation algorithm, as is, leads to significantly better performance. In Fig. 5 only one simple example is analyzed. Also, the results from Fig. 4 seem inconclusive. It would be good to report the number of iterations it takes for the continuation algorithm to converge.

Originality: The idea of re-explaining regularization using orbitopes is new. The idea of a continuation algorithm that iteratively morphs the shape of ball-norms in addition to scaling them is, as far as I can tell, new.
Summary: The paper is well written and has no technical mistakes but most of the contributions consist of re-explaining previously introduced ideas in a different language (orbitopes).

Their continuation algorithm is the contribution that most clearly allows one to actually do something in a different way (maybe leading to improved solutions over other algorithms) but, unfortunately, the algorithm comes with no guarantees and the numerical results are a bit lacking.

Submitted by Assigned_Reviewer_26

This paper provides a new insight on sparsity inducing regularizations. The authors imported the notion of group-majorization and showed that several well-known regularizations are recovered by properly introducing a group G and a seed vector v. The authors also provided gradient-based optimization methods and a regularization path heuristic.

Overall, the paper is well-written and the main idea is clear. Characterization of sparsity inducing regularizations is important since sparsity is fundamental in recent machine learning, and this work would help us for deeper understanding.

The use of the orbitope would be an unique point of this research. However, the discussion on its utility seems not sufficient. As stated in Corollary 4, the orbitope and the atomic norm are relevant, and the premutahedra and sorted l1-norms discussed in the paper are both atomic norms. The orbitope and the atomic norm may be different in general, but how does this difference brings us to a new insight? In particular, can we find practically useful new regularizations with a help of the orbitope, in which any existing studies could not? Further discussion on this point will be beneficial. I think the regularization path heuristic in Section 6 would be one advantage of the orbitope. This heuristic allows us to adaptively tune the regularization, which will not be available with the atomic norm.
Summary: This research provides a new way to interpret sparsity inducing regularizations using the orbitope. The content is interesting, although the advantage of the use of the orbitope is yet unclear.

Submitted by Assigned_Reviewer_41

The paper investigates a group-theoretic view of penalized likelihood functions in vector spaces. It is shown that many commonly used regularization terms, such as the L_1-, L_2-, and L_infty-norm, are specific instantiations of what is here called "orbit regularization". The generic view is shown to suggest also new reasonable forms of regularization, which admit nice properties when it comes to optimization using conditional and projected gradient algorithms.

Quality:

The work is of very good quality. The relevant literature, both old and recent, is well cited.

Clarity:

The presentation is excellent. The notation is carefully chosen and adheres the usual conventions. The definitions are clear. The paper is pleasant and relatively easy to read, even if the subject is quite abstract.

The paper does not adhere to the NIPS section headings and referencing styles.
Line 035: "klowledge". In Def 5, Prop 6, etc., consider placing the period (in bold) outside the parentheses, or just remove it.

Originality:

It seems that the proposed view is new to the machine learning community. However, much of the underlying mathematics does not look that new: for example, Propositions 11 and 12 are attributed to earlier works as old as Eaton (1984) and Hardy et al. (1952). The paper could better crystallize the its contributions.

Significance:

The significance of the work is unclear. In particular, it is not clear how the work advances the state of the art when it comes to the practice of machine learning. In general, it is nice to have several computationally efficient forms of regularization available. On the other hard, there is no objective way to select "the best" among them for a given learning problem. So, just generating new forms will not lead to very significant advancements in the field.
Summary: Well presented, carefully typed, view that may contribute to better understanding of likelihood regularization in "linearly flavored" machine learning problems.
Author Feedback
Author rebuttal: We would like to thank all reviewers for their insightful comments.

R1 and R2 ask about the practical benefits of orbitopes and the new insights they bring. While orbitopes and atomic norms are closely related (as shown in Sec. 2.4), the latter do not come with any guidance for choosing the set of atoms. By contrast, orbitope regularizers are generated by a group structure and a seed -- this decouples the "invariance" properties of the regularizer (given by the group) from the actual shape of the ball (given by the seed). We believe this is a very natural way to regard regularization. This decoupling is not only conceptual, it's also key for the regularization path heuristic in Sec. 6 -- correctly hinted by R1 as one important advantage of orbitopes. Another advantage is the ability to design problem-specific regularizers based on their intended invariances (dictated by the choice of group). Different atomic norms spanned by the same group can be handled computationally in a unified manner via group-centric properties (see Secs. 3 and 5). We will highlight these advantages in the final version.

We agree with R1's suggestion of adding further discussion on the ability "to find practically useful new regularizations with a help of the orbitope." We regard our paper as a first step toward group-based regularization, and the regularizers studied here may be just the tip of the iceberg. Groups are well studied and classified in other disciplines, and chances are high that this framework could inspire new regularizers that are a good fit to specific ML problems. Our paper focuses on setting up the key ingredients and pinpointing the properties that make groups amenable to be plugged into existing proximal-gradient and Frank-Wolfe learning algorithms.

In a similar vein, we will "better crystallize [our paper's] contributions," as R2 suggests. While the underlying mathematics (orbitopes and majorization theory) is not our own contribution, these concepts have never been used in ML (to the best of our knowledge). We make a bridge between that line of work and regularization in ML (with several examples in Sec. 2); and we present several novel results, such as deriving the properties of groups that can be exploited by optimization algorithms (Secs. 3 and 5), unveiling the relation with sorted-L1 norms (Sec. 4), and the derivation of the continuation algorithm (Sec. 6).

R3 has a few technical questions about the continuation algorithm, which we answer here briefly (full details will be added to the final version). Regarding "convergence": (1) for any epsilon > 0 the sequence {L(w_t)} is strictly decreasing and convergent -- see proof sketch below. (2) for large enough epsilon, the algorithm is guaranteed to stop after a finite number of steps. Intuitively, the balls keep increasing until they reach a radius above which the regularization constraint ceases to be active. At this point, the optimal w_t will lie strictly inside the ball (this is what we mean by "at the point where regularization is not having any effect.") The principle is the same as in classical homotopy methods, but our case does not lead to a "complete" path (we're not just increasing the ball but also exploring its shape, via the heuristic in line 5 of Fig. 3). The question of "how the algorithm would perform when epsilon = 0" is interesting. In this setting, the size of the ball would be fixed and only its shape would change. However, the algorithm would no longer be tracing a regularization path. Unless the noise level is known, further search would be necessary to pick the ball size.

R3 says "our numerical results do not clearly show that the continuation algorithm leads to significantly better performance." While we agree this is a toy experiment, we believe the results are encouraging and provide a proof of concept. Fig. 5 (right) shows that the continuation method (both for signed and unsigned permutahedra) outperformed the L1 baseline by a significant margin: we have around 0.037 averaged test error for L1 against 0.031 and 0.028 for the permutahedra with the continuation method (after cross-validation). Similar results were obtained for other choices of n and k (omitted for space). The average number of iterations was below 100, and we set epsilon=0.1. We will add this missing information and clarify the experiments.

We will fix the minor formatting issues pointed out by R2 and R3, including the y-axis in Fig. 4. In particular, we acknowledge R3's suggestion for the pictorial explanation of Proposition 10.

--

Erratum: in Fig. 3, line 5, v_t should be replaced by v_t' \in Gv_t \cap K_G(w_t) -- i.e., we pick a v_t' in the orbit of v_t that is "maximally aligned" with w_t (our implementation is correct).

Proof sketch (convergence of the continuation algorithm): we show that for any epsilon > 0, the sequence {L(w_t)} is strictly decreasing. Convergence follows from the fact that it is lower bounded by the unregularized objective value. The proof consists of two steps:

(1) Showing that, for any epsilon>0, w_t lies in the interior of O_G(v_{t+1}). This follows from the fact that v_t', w_t, and their convex combination all belong to the region cone K_G(w_t); in this region the pre-order induced by G is a cone ordering w.r.t. the polar cone of K_G, from which we can derive w_t \in O_G(alpha*v_t' + (1-alpha)*w_t), leading to the desired statement.

(2) Showing that {L(w_t)} strictly decreases before the algorithm terminates. This is a simple consequence of (1). Since w_t \in O_G(v_{t+1}), we must have L(w_{t+1}) <= L(w_t). If this holds with equality, then w_{t+1}=w_t is an optimal solution at the (t+1)-th iteration, but since it lies in the interior of O_G(v_{t+1}), we have ||w_{t+1}||_{Gv_{t+1}} < 1 and the algorithm will terminate. Therefore we must have L(w_{t+1}) < L(w_t) for the algorithm to proceed.